# Malaria preventive practices and delivery outcomes: A cross-sectional study of parturient women in a tertiary hospital in Eastern Uganda

**Rebecca Nekaka[1], Julius Nteziyaremye[2], Paul Oboth[1], Jacob Stanley Iramiot[3]\***, Julius Wandabwa[2]

1 Department of Community and Public Health, Busitema University Faculty of Health Sciences, Tororo, Uganda, 2 Department of Obstetrics and Gynecology, Busitema University Faculty of Health Sciences, Tororo, Uganda, 3 Department of Microbiology and Immunology, Busitema University Faculty of Health Sciences, Tororo, Uganda

☯ These authors contributed equally to this work.
* jiramiot@gmail.com

**Data Availability Statement:** All relevant data are within the manuscript and its Supporting Information files.

## Abstract

### Background

Uganda ranks third in the number of deaths attributable to malaria and has some of the highest recorded malaria transmission rates in the general population. Malaria in Pregnancy is associated with detrimental effects for the mother and unborn baby and these effects seem to have long term effects and consequences on the life of the baby. Despite the preventive measures put in place by the World Health Organization in antenatal care, the burden of malaria in pregnancy is still high. We determined the use of malaria preventive strategies during pregnancy and the presence of plasmodium infection, anemia, and low birth weight babies at delivery among parturient women at Mbale regional referral hospital in eastern Uganda.

### Methods

A cross-sectional study was conducted among 210 women delivering at MRRH between July 2017 and January 2018. Information on demographics, antenatal care, and prevention practices was collected using an interviewer-administered questionnaire. Maternal venous blood and cord blood samples were screened for Plasmodium infection by both microscopy of Giemsa-stained blood films and *Plasmodium falciparum* rapid diagnostic test (pf. HPR2 mRDT). Polymerase Chain Reaction (PCR) was done on cord blood. The presence of anemia was determined by the use of an automated hemoglobin analyzer. Data were analyzed using descriptive and analytical statistics.

### Results

Of the 210 women, 3 (1.4%) and 19(9.1%) tested positive for malaria by using Giemsa stained blood smear microscopy and malaria rapid diagnosticMRDT tests respectively. PCR detected 4(%) of Plasmodium in cord blood. Twenty-nine percent of the women had anaemia and 11 (5.2%) had low birth weight babies. Only 23.3% of the women received at

**Funding:** This study was supported by the Makerere University-Swedish International Development Agency (Mak-SIDA) Research Cooperation Fund with Public Partner Universities in Uganda. The funders had no role in study design, data collection and analysis, decision to publish, or preparation of the manuscript.

**Competing interests:** The authors have declared that no competing interests exist.

**Abbreviations:** ANC, Antenatal Care; DOT, Directly Observed Treatment; IPT, Intermittent Preventive Treatment; IPTp-SP, Preventive Treatment with Sulfadoxine- Pyrimethamine; ITN, Insecticide Treated Net; LBW, Low Birth Weight; MIP, Malaria in Pregnancy; MOH, Ministry of Health; MRRH, Mbale Regional Referral Hospital; mRDT, Rapid Diagnostic Test; UDHS, Uganda Demographic Health Survey; WHO, World Health Organization.

least three doses of IPTp-SP and 57.9% reported sleeping under an Insecticide Treated Net the night before the survey. The women who reported sleeping under a mosquito net the previous night (OR 0.67, 95% CI: 0.24–1.86) and those who reported taking fansidar as a directly observed therapy (OR 0.31, 95% CI: 0.04–2.39) appeared to have few chances of getting plasmodium infection though the findings were not statistically significant.

## Conclusion

The effective use of malaria preventive strategies (IPT-SP and Insecticide Treated Nets) was generally low. Most of the women took less than three doses of SP and there was no strict adherence to the recommended directly observed therapy. The prevalence of Plasmodium infection during pregnancy was low though maternal anaemia and low birth weight were relatively high.

## Background

Africa carries a disproportionately high burden of malaria globally [1]. In 2008, Uganda ranked third in the number of deaths attributable to malaria and has some of the highest recorded malaria transmission rates in the general population [2]. Pregnant women have been reported to have three times more likelihood to suffer from severe forms of the disease than their non-pregnant colleagues [3]. Consequences of malaria in pregnancy include but not limited to premature delivery, intrauterine fetal demise, low birth weight neonates, and neonatal death [4–7]. Severe anaemia and maternal death are also more expected to occur among pregnant women than their non-pregnant colleagues [8]. The mortality rate from severe malaria approaches 50% [3, 4] depending on several factors ranging from the woman's immune status to the existence of comorbid conditions, trimester of pregnancy, and gravidity [9]. In a bid to reduce malaria burden in pregnancy, the World Health Organization (WHO) endorsed a set of interventions in areas with the stable (high) transmission of *P. falciparum*. These interventions include the use of insecticide-treated nets (ITNs), intermittent preventive treatment with sulfadoxine-pyrimethamine (IPTp-SP), and effective case management of malaria illness and anaemia [10, 11]. Despite the adoption of the WHO strategy, there are still reports of poor uptake of malaria preventive measures. The percentage of women who slept under insecticide-treated nets (ITNs) was reported to be 64% and only 17% of the women reported taking three or more doses of IPT- SP [12].

The IPT as well as ITNs are made available by the government and provided free to pregnant mothers during ANC visits. However, data quantifying the impact of the preventive measures on malaria in pregnancy and birth outcomes are limited. Despite the availability of these preventive measures, the burden of malaria among pregnant women is still high (prevalence). To monitor the impact of the preventive practices, there is a need to evaluate and have information on its implementation. This study determined the use of malaria preventive strategies during pregnancy and the presence of plasmodium infection, anemia, and LBW among parturient women at Mbale regional referral hospital in eastern Uganda.

## Materials and methods

### Study site

The study was conducted from July 2017 to January 2018 in the Obstetrics and Gynecology department at Mbale Regional Referral Hospital (MRRH). The hospital is a tertiary health care

facility located in the eastern part of Uganda and serves the districts of Mbale, Manafwa, Bududa, Busia, Budaka, Kibuku, Bukwo, Butaleja, Pallisa, Sironko, Bulambuli, Kween, and Tororo, including several lower-level Government-run health centers, private not for profit health units and patients outside its catchment area. MRRH is a teaching hospital for Busitema University Faculty of Health Sciences and also serves as an internship center in Uganda. The Hospital has a 355-bed capacity and the department of obstetrics and gynecology conducts annual deliveries of about 10,014. Eastern Uganda is a malaria-endemic area with transmission all the year round and peaks during the rainy season. Mbale town, where MRRH is located is traversed by Manafwa River that provides a breeding ground for mosquitoes. The total population of this area is 492,804.

### Study design and population

This was an analytical cross-sectional study.

The study population was women giving birth at Mbale regional referral hospital.

### Sampling procedure

Study participants were recruited consecutively as they presented during labor. On average five to eight participants were recruited per day and interviewed by trained assistants. Informed oral and written consent was obtained after explaining the study and those who declined enrolment were excluded from the study.

### Inclusion / exclusion criteria

The women who attended the antenatal clinic and presented in labor at term (between 37 and 40 weeks of gestation) were included in the study. Those who had no supporting records for ANC and women who had taken malaria prophylaxis other than IPTp-SP were excluded.

### Sample size calculations

The study's minimum sample size was 184. This was calculated using Kish Leslie formula (1965) with a 13.9% prevalence of malaria parasitemia at delivery [13] with a precision of 5% and a standard normal deviation of 1.96 at 95% confidence intervals. Considering a nonresponse rate of 10% and sample attrition, data were collected from 210 participants.

### Data collection techniques

A pre-tested structured questionnaire was administered by trained research assistants and they collected information on demographics, malaria preventive strategies used, doses and trimester of administration of IPTp-SP, and haemoglobin levels during the ANC visits and at delivery. Information on IPT use, previous Plasmodium infection, and malaria treatment during pregnancy were extracted from the ANC records.

Baby's birth weight was collected from the labour ward record. LBW babies were categorized as those with birth weight <2500 g.

### Sample collection and laboratory tests for malaria parasite and anemia

Trained laboratory scientists collected 5mls of venous blood from the mother using a vacutainer system and 2mls of cord blood from the newborn into separate EDTA bottles from each study participant. Venous blood was collected before delivery for peripheral blood diagnosis of malaria and hemoglobin testing. Each sample was given a number and paired (mother-newborn) and labeled with the participant's information. Thick and thin blood films were made

from both venous and cord blood, then stained with Giemsa. Peripheral parasitemia was determined by the number of parasites per 200 white blood cells. Slides without parasites were indicated as; No Malaria Parasites Seen (NMPS). All positive and negative slides were subjected to a second blinded reader with over 5 years' experience in malaria microscopy from Mbale regional referral hospital. A third reader, who is a WHO certified microscopist at Mbale Regional Referral Hospital read slides that had discordant readings by the first two readers. Each time a thick smear was prepared on maternal venous blood, an mRDT was also done using the *P. falciparum* specific antigen HRP2 type. Results for mRDT were recorded as either positive or negative [14]. There were no invalid test results.

**Polymerase Chain Reaction (PCR) detection and speciation of Plasmodium.** Plasmodium species were detected and speciated in dried blood spot samples using Genus-specific Polymerase Chain Reaction (PCR) followed by a nested species-specific PCR of 18S small subunit ribosomal DNA. Species were determined based on product size on 2.5% agarose gel. Briefly, DNA was isolated from dried blood spots (DBS) by using the Chelex extraction method as previously described template DNA was amplified using nested PCR, with second-round primers specific to the species *Falciparum*, *Vivax*, *Malariae*, and *Ovale*. Separate reactions were performed for each pair of nested primers. For example, one 96-well plate of primary PCR became four 96-well plates of each species if all four are to be determined. A No Template Control (NTC) was used in all reactions and genomic DNA from laboratory strains or clinical isolates was used as a positive control for respective species. Gel electrophoresis was done and digital images were taken in an imaging cabinet under UV light. Gel images were printed and corresponding sample lanes were scored visually for the presence of specific species. Since each nested PCR is specific to only one species if present, only the corresponding amplified products appeared in the gel lane.

## Primers for both rounds of PCR are listed below

| Genus | Primer Name | Sequence |
|---|---|---|
| **Plasmodium** | rPLUf | 5'– TTA AAA TTG TTG CAG TTA AAA CG |
| | rPLUr | 5'– CCT GTT GTT GCC TTA AAC TTC |
| **Nested round:** | | |
| **Falciparum** | rFALf | 5'– TTA AAC TGG TTT GGG AAA ACC AAA TAT ATT |
| | rFALr | 5'– ACA CAA TGA ACT CAA TCA TGA CTA CCC GTC |
| **Vivax** | rVIVf | 5'– CGC TTC TAG CTT AAT CCA CAT AAC TGA TAC |
| | rVIVr | 5'– ACT TCC AAG CCG AAG CAA AGA AAG TCC TTA |
| **Malariae** | rMALf | 5'– ATA ACA TAG TTG TAC GTT AAG AAT AAC CGC |
| | rMALr | 5'-AAA ATT CCC ATG CAT AAA AAA TTA TAC AAA |
| **Ovale** | rOVAf | 5'– ATC TCT TTT GCT ATT TTT TAG TAT TGG AGA |
| | rOVAr | 5'– GGA AAA GGA CAC ATT AAT TGT ATC CTA GTG |

**Primary round.** The hemoglobin level was determined using an automated hematology analyzer. The presence of anemia was considered as Hb less than 11 g/dl [15].

## Data management and analysis

The questionnaires used to collect data for the study were serial numbered for data entry. The data were double entered into Microsoft Excel, cleaned, and thereafter imported to STATA software version14 for analysis. Data were presented using descriptive statistics for all variables; a chi-square test was used to determine the associations between delivery outcomes

(maternal Plasmodium infection, maternal anemia, and birth weight) and malarial preventive strategies.

The response variables were all binary outcomes therefore, the first step of the model building process was to perform univariate analysis to identify important predictors between the independent and dependent variables using a simple logistic regression model for each outcome(s). All variable(s) showing at least a moderate association with each response were selected by considering the p-value less than 25% as Hosmer and Lemshow recommended [16]. Also, variables that have shown to be significant in other studies or are of biological importance were selected to be included in the final model. Secondly, the multiple logistic regression model was fitted using variables selected in the first step. Backward stepwise elimination selection procedures were used to select the most important variables that were significantly associated with each outcome(s). The association between each preventive measure used and the outcome was determined using multivariate logistic regression while controlling for the others.

## Ethics approval and consent to participate

Ethical approval for the study was obtained from Mbale Regional Referral Hospital research and ethics committee, Uganda (COM 058/2017). Each participant provided written informed consent before enrolment to the study. Mothers with parasitemia and anemia were referred to the obstetricians for further management. Information collected from the participants was kept confidential and stored in hard-locked cabinets.

## Results

The age of mothers ranged from 14 to 43 years with a mean age of 24.63 years (SD ± 6.45years) at the time of childbirth. The majority (58.1%) of the mothers were below 25 years of age. Approximately 81% of the mothers were earning less than 300,000ugshs per month. More than half of the mothers 51.90% (n = 109) had attained secondary and tertiary level education, a factor that may influence health-seeking behavior. Fifty-nine percent (n = 124) were either prime or second gravid. A majority of the mothers 88.57% (n = 186) were married (**Table 1**).

As regards antenatal care attendance, 49.5% had attended at least 4 contacts (**Fig 1**).

## Utilization of preventive measures by the women during antenatal period

Of the 191 women who reported receiving IPT with sulfadoxine-pyrimethamine (IPTp-SP) during antenatal care, the majority (n = 142) received less than three doses of IPT-Sp and 14.3% (n = 30) received IPT as directly observed therapy (DOT). The majority 85.7% (n = 180) of the women reported receiving IPT with sulfadoxine-pyrimethamine (IPTp- SP) as a prescription and not as DOT. Slightly more than half of the mothers, 57.9% (n = 121) mentioned they slept under a mosquito net the night preceding the interview (**Table 2**). More than half of the women, 145 (69.1%) had a malaria test done during antenatal care and among these, 39 (26.9%) had a positive test for malaria. Most of the women (97.1%) received iron/ folate supplementation during ANC.

## Prevalence of malaria (maternal and neonatal), maternal anaemia and low birth weight among the parturient women

Of the 210 parturient women, 1.4% (3/210) and 9.1% and (19/210) tested positive for malaria using slide microscopy and malaria pf (HRP2) Ag mRDT (Plasmodium *falciparum* histidine-rich protein 2 antigen rapid diagnostic Test) tests respectively. Of the 210 matched cord blood

**Table 1. Demographic characteristics and delivery outcomes.**

| Demographics characteristics | Frequency N = 210 (%) | mRDT+n (%) | ANEMIA n (%) | LWB n (%) |
|---|---|---|---|---|
| **Age group** | | | | |
| 14–24 | 122(58.1) | 16(84.2) | 42(68.9) | 8(72.7) |
| 25–29 | 39(18.57) | 2(10.5) | 9(14.8) | 2(18.6) |
| > = 30 | 49(23.33) | 1(5.3) | 10(16.4) | 1(9.1) |
| **Education level** | | | | |
| None | 13(6.2) | 0(0.0) | 2(3.3) | 0(0.0) |
| Primary | 88(41.9) | 10(52.6) | 32(52.5) | 9(81.8) |
| Secondary | 86(41.0) | 6(31.6) | 22(36.1) | 2(18.2) |
| Tertiary | 23(11.0) | 3(15.6) | 5(8.2) | 0(0.0) |
| **Monthly Income** | | | | |
| 300,000/ = and Above | 39(18.6) | 4(21.1) | 8(13.1) | 2(18.2) |
| < 300,000/ = | 171(81.4) | 15(78.9) | 53(86.9) | 9(81.8) |
| **Marital Status** | | | | |
| Divorced | 5(2.4) | 0(0) | 2(3.3) | 0(0) |
| Married | 186(88.6) | 18(94.7) | 52(85.3) | 8(72.7) |
| Single | 19(9.1) | 1(5.3) | 7(11.5) | 3(27.3) |
| **Parity** | | | | |
| Prime gravida | 84(40.0) | 8(42.1) | 28(45.9) | 5(45.5) |
| Second gravida | 40(19.1) | 8(42.1) | 10(16.4) | 1(9.1) |
| Multi-gravida | 86(41.0) | 3(15.8) | 23(37.7) | 5(45.5) |

samples, 2 tested positive using slide microscopy, 0.95%. The prevalence of Plasmodium in cord blood as detected by PCR was 4%. On speciation, *Plasmodium falciparum* was the only species detected. This means early neonatal malaria prevalence was about 1%. The majority,

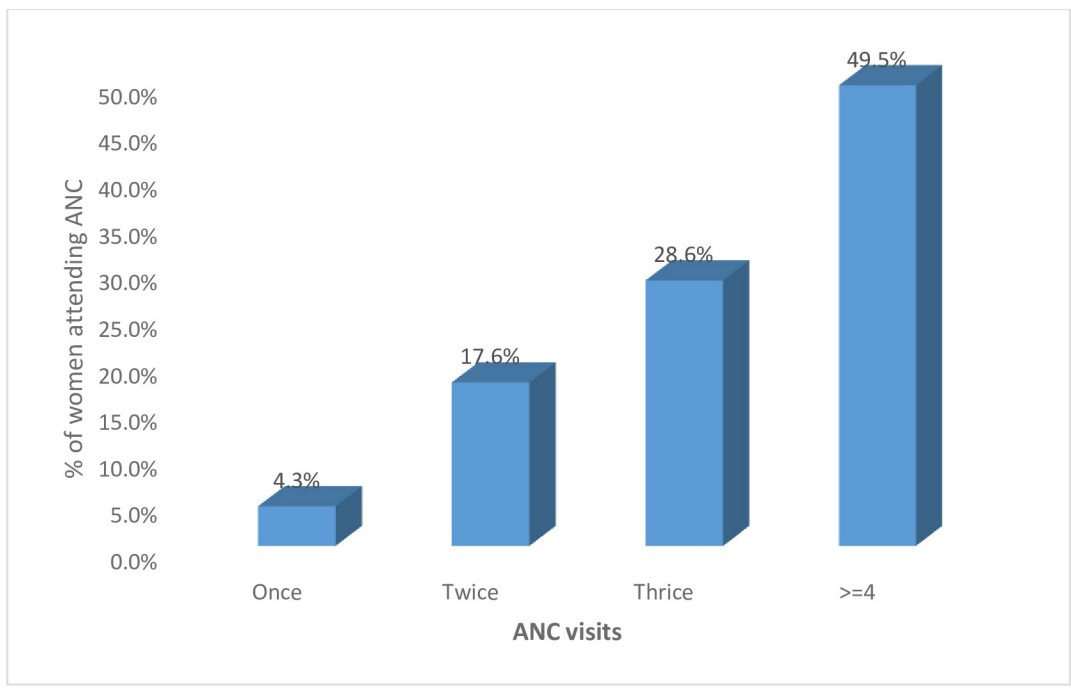

**Fig 1. Antenatal attendance.**

**Table 2. Malaria preventive measures and the delivery outcomes.**

| Preventive factors | Frequency N = 210(%) | mRDT+n = 19 (%) | ANEMIA n = 61 (%) | LWB n = 11 (%) |
|---|---|---|---|---|
| *Doses of Fansidar taken* | | | | |
| *None* | 19(9.1) | 1(5.3) | 2(3.3) | 3(8.0) |
| *1–2 doses* | 142(67.6) | 13(58.4) | 43(70.5) | 4(36.4) |
| *> = 3 doses* | 49(23.3) | 5(26.3) | 16(26.2) | 4(36.4) |
| *Method of delivery of fansidar* | | | | |
| *DOT* | 30(14.3) | 1(5.3) | 7(11.5) | 1(9.1) |
| *Prescription* | 180(85.7) | 18(94.7) | 54(88.5) | 10(90.9) |
| *slept in ITN the previous night* | | | | |
| *No* | 89(42.1) | 10(52.6) | 26(42.6) | 6(54.6) |
| *Yes* | 121(57.9) | 9(47.4) | 35(57.4) | 5(45.4) |

84% (16/19) of the mothers with positive malaria Rapid Diagnostic Test (mRDT) at delivery were young mothers below 25years of age, with a low level of education (52.6% stopped at the primary level of education), low monthly income (78.9% had a monthly income less than three hundred thousand shillings) and majority were paucigravida. Anaemia in pregnancy defined as a haemoglobin concentration below or equal to 10.9g/dl was very prevalent at 29% (61/210). The majority were young mothers: 68.9% were below 25 years old. Slightly above half, 52.5% had stopped at the primary level, 86.9% had a low income while and 62.3% were paucigravida. The low birth weight prevalence amongst these mothers was 5.2% (11/210). The majority, 72.7% were below 25 years of age and 81.8% had stopped at primary school level of education and were low-income earners earned less than three hundred thousand Uganda shillings per month.

## Malaria preventive measures and delivery outcomes

Among the mothers who tested positive for malaria using mRDT, only 26.3% (n = 5) had taken up three or more IPTp-SP doses while 73.4% had had less than 3 IPTp-SP. Of these, 94.7% (n = 18) received IPT-SP by prescription. Slightly more than half (52.6%) of the women with a positive mRDT had not slept in a mosquito net the previous night before the survey.

Among the 61 women who had anaemia, 43 (70.5%) had received only 1–2 doses of IPTp-SP and 88.5% received it by prescription. Moreover, 90% of the women who had low birth weight babies received IPTp-SP by prescription as opposed to the recommended DOT (**Table 2**).

In the univariate logistic regression analysis, the method of IPTp-SP administration by directly observed therapy (OR 0.31, 95% CI: 0.04–2.4) and sleeping under a mosquito net the previous night (OR: 0.63, 95%CI: 0.25–1.63) appeared to be protective against maternal plasmodium infection although the findings did not show statistical significance (**Table 3**).

From the literature, method of delivery, sleeping under a net, and doses of sulfadoxine-pyrimethamine (SP) have been shown to predict maternal plasmodium infection hence they were analyzed in the multivariable logistic model (**Table 4**). The women who slept under a mosquito net the previous night (OR 0.67, 95% CI: 0.24–1.86) and those who took SP as a directly observed therapy (OR 0.31, 95% CI: 0.04–2.39) appeared to have fewer chances of getting plasmodium infection although the findings did not show statistical significance.

There was no significant association between maternal Plasmodium infection and maternal anaemia. Being of age thirty and above was protective against anaemia. The women who took 2 or more doses had a higher likelihood of being compared to those who had less than two doses.

**Table 3. Factors associated with maternal Plasmodium infection.**

| Variables | Odds ratios (95% CI) | P-value | Adjusted OR(95% CI) | p-value |
|---|---|---|---|---|
| *Demographics* | | | | |
| *Education level (primary as reference)* | | | | |
| Secondary and above | 0.82 (0.32–2.10) | 0.679 | | |
| *Household Income* | | | | |
| > = 300,000 UGX | 1.19 (0.37–3.80) | 0.771 | | |
| *Parity (prime as reference)* | | | | |
| Second | 2.37 (0.82–6.87) | 0.111 | 2.67 (0.88–8.06) | 0.082 |
| Multiparty | 0.34 (0.08–1.34) | 0.124 | 0.85 (0.16–4.35) | 0.844 |
| *Age groups (years) (14–24, as reference)* | | | | |
| 25–29 | 0.36 (0.78–1.63) | 0.185 | 0.43 (0.08–2.24) | 0.314 |
| > = 30 | 0.13 (0.18–1.07) | 0.058 | 0.21 (0.02–2.24) | 0.195 |
| *Preventative strategies* | | | | |
| *Net Use (No as reference)* | | | | |
| Yes | 0.63 (0.25–1.63) | 0.346 | 0.67 (0.24–1.86) | 0.445 |
| *Fansidar doses taken (None as reference)* | | | | |
| 1–2 doses | 1.81 (0.22–14.71) | 0.577 | 1.88 (0.21–16.6) | 0.569 |
| > = 3 doses | 2.04 (0.22–18.76) | 0.527 | 1.78 (0.17–18.3) | 0.625 |
| *Fansidar method of delivery (prescription as ref.)* | | | | |
| DOT | 0.31 (0.04–2.41) | 0.264 | 0.30 (0.04–2.39) | 0.255 |

# Discussion, limitations, conclusion, and recommendation

## Discussion

Overall malaria prevalence using peripheral smear was 1.4%. This was lower than the 9% reported a decade ago by Namusoke et al in Mulago and 29.7% by Filbert J Mpogoro et al in Tanzania [13, 17]. In the Tanzania study by Mpogoro et al, the prevalence by mRDT was lower

**Table 4. Factors associated with maternal malaria.**

| Variables | Odds ratios (95% CI) | P-value | Adjusted Odds ratios (95% CI) | P-value |
|---|---|---|---|---|
| *Demographics* | | | | |
| *Education level* [Reference: < = primary] | | | | |
| Secondary and above | 0.65 (0.36–1.18) | 0.157 | 0.55 (0.29–1.06) | 0.073 |
| *Household Income* [Reference: < 300,000 UGX] | | | | |
| > = 300,000 UGX | 0.57 (0.25–1.33) | 0.197 | | |
| *Age groups (years)* [reference: 14–25] | | | | |
| 25–29 | 0.57 (0.25–1.31) | 0.188 | 0.65 (0.27–1.54) | 0.330 |
| > = 30 | 0.49 (0.22–1.07) | 0.075 | 0.43 (0.19–0.98) | 0.045 |
| *Preventative strategies* | | | | |
| *IPT doses* [Reference: < 2 doses] | | | | |
| > = 2 doses | 2.00 (0.97–4.10) | 0.058 | 2.15 (1.03–4.52) | 0.041 |
| *mRDT* [Reference: No] | | | | |
| Yes | 1.48 (0.55–3.96) | 0.435 | | |
| *Malaria ANC* [Reference: Negative] | | | | |
| Positive | 0.95 (0.42–2.14) | 0.903 | | |
| *Iron folate* [Reference: No] | | | | |
| Yes | 0.81 (0.14–4.56) | 0.815 | | |

than that by peripheral smear, that is,19.5% versus 29.7% [17] in contrast to ours that was 9.1% by mRDT and 1.4% by peripheral smear. This could be explained by our population selection in which we focused on those that had evidence of having attended ANC whereas other studies included all the women that had delivered in their study areas. This could also be explained by the change in lifestyle and efforts that have been scaled up to curb malaria in pregnancy since 2013 [18]. The low prevalence finding in our study based on microscopy could be due to the study site despite being a high transmission zone experienced a prolonged dry season and the study was done just at the start of the rainy season and this may explain the low prevalence of plasmodium infection. The increase in antenatal attendance and improved uptake of IPTp may also be responsible for the low prevalence of malaria in pregnancy in this region. Looking at the discrepancy between microscopy results and mRDT it comes out clearly that pregnancy poses specific challenges for the diagnosis of malaria. *P. falciparum* parasites may be present in the placenta but absent or undetectable in peripheral blood. The high prevalence of 9.1% by mRDT in this study could be associated with the persistence of HRP2 antigen or residual anti-genaemia in peripheral blood, leading to positive mRDTs in the absence of parasitaemia. The difference in the prevalence between microscopy and mRDT results in this study is based on the limitations in the sensitivity of the two methods, therefore it is necessary to use more superior methods like histology which provides insight on pathological changes as well as the timing of infection (acute, chronic and past) [13]. The mRDTs have also been reported to be more reliable in the detection of malaria in pregnancy than slide microscopy [19]. In another study, 7.83% of asymptomatic gravid mothers had malaria while 17.97% of symptomatic cases had malaria [7]. The prevalence of malaria among asymptomatic gravid mothers continues to show that we could have missed past infections among such women since we relied solely on blood film examination only and never took the placental samples [14].

Studies elsewhere in sub-Saharan Africa have reported prevalence as high as 40.0% [7].

Plasmodium infection among women below 25 years was higher than those above 25 years. This finding is consistent with findings from a study in Mulago where higher infection rates were found in the young primigravidas [13] and similar findings were reported in a study done in Nigeria [9].

Overall, only 23.3% of parturient women received at least three doses of IPTp-SP. Although low, this was above the national level of 17% [12]. Uptake of IPTp-SP seems to be a challenge for pregnant women and this could be because low levels of ANC contacts are realized. In a study in Zambia to evaluate IPTp-SP and retrospective birth outcomes in Mansa, Zambia 67% of the women had had no or only two doses [20]. Another study in Malawi showed that only 47% of mothers had at least 4 ANC contacts and only 52% received optimal SP doses [21]. In a study in Tanzania, only 6% of the women received three or more doses [17] while 47.7% of the women in their third trimester in Ghana received less than three doses and 35% received only one dose [22]. Uganda's national policy for IPTp administration during pregnancy recommends early initiation of IPTp in the second trimester (at 13 weeks of gestation) for all women. Timely uptake of IPTp depends more on the practices of health workers at the health units coupled with quality improvement geared at improving services with accompanying strong leadership than individual characteristics of pregnant women [23]. The lower than expected uptake of IPT in our study could reflect suboptimal care at the antenatal clinics, possible stock out of drugs in the health units, and inadequate sensitization of health workers about the new guidelines on IPTp administration to pregnant women. This could be supported by the fact that 18 of the 19 mothers that tested positive using mRDT had had SP by prescription rather than Directly observed treatment (DOT) and probably had not bought the drugs.

Regarding ITN use, 65% of the overall household population in Uganda has access to an ITN [12] and the survey further highlights that 64% 0f the pregnant women slept under ITN

the night before the survey. However, in our study, only 57.9% of the mothers had slept under ITN the night before the survey which is below the national level and lower than 67.3% reported by Namusoke et al [13] but higher than that reported in Cameroon of 32.4% [24]. The low bed net use in our study could be due to knowledge gaps among the women on the importance of the nets in preventing malaria even when the government had put in huge amounts of funds to purchase and distribute nets.

All of the 210 women attended ANC but only 49.5% of them attended four times or more as recommended by the World Health Organisation. This level of ANC attendance is comparable to the regional levels of 47% [12].

In our study, the prevalence of anaemia was high (29.1%) despite the 97% folate/iron supplementation during pregnancy. Our prevalence was higher than findings from a study in Mulago where the prevalence was 22% [13]. Being of age 30 years and above was protective against anaemia (*P<0.05*) though Plasmodium infection at delivery was not significantly associated with maternal anaemia. In contrast, maternal anaemia was significantly associated with Plasmodium infection in a similar study in Benin [25]. In the tropics, anaemia is multifactorial and may be caused by parasitic infections, haemoglobinopathies, and poor nutrition among other factors. A similar study in Mpigi, Uganda associated the high prevalence of maternal anaemia to Plasmodium infection, HIV seropositivity lack of both IPT and iron supplementation [26]. The potential role of other infections other than malaria was not assessed in this study.

The prevalence of LBW in this study (5.2%) was lower than that recorded in a study in Mulago [13] where the prevalence was 12.2%. Most of the LBW babies in our study were born to mothers who had taken less than three doses of IPT and mostly Primigravidae. Other findings from studies elsewhere showed a prevalence of LBW as high as 37% and mostly in those below 25years [9, 24].

From the multivariate analysis, taking two or more doses was associated with a higher prevalence of maternal anaemia in our study. This finding is in contrast to a study in Malawi where IPT was associated with an improved delivery outcome [27]. In Fort Portal, Western Uganda, [28] a study reported that the use of IPTp-SP did not show observable benefit. In Tororo, Eastern Uganda: an area with higher malaria endemicity compared to Western Uganda, a high prevalence of placental Plasmodium infection was reported among women who had had at least 2 doses of SP during pregnancy and the use of ≥2 doses of SP was not associated with protection against individual birth and maternal outcome measures but did protect against a composite measure of any adverse outcome [29]. The results of a study in Tororo (Uganda) and others suggest the efficacy of IPT with SP may be reducing in East Africa due to the development of drug resistance hence WHO has endorsed increasing the number of doses to increase efficacy [18]. In our study, most of the women took 1–2 doses of SP as opposed to the recommended three or more doses.

The number of IPT doses taken was not significantly associated with maternal Plasmodium infection in this study. These findings are different from a study in Tanzania where the uptake of three or more doses of IPT was associated with reduced prevalence of placental malaria [17]. Namusoke in her study in Mulago found that women who reported to have taken fansidar had not taken it and had low levels of the drug in their blood [13]. The lack of association in our study could have been due to misclassification since these were self -reports and abstraction from ANC records and not directly observed therapy.

In our study, there was a 4% prevalence of congenital malaria. Congenital malaria has been reported before in Eastern Uganda, particularly in Mbale [30] and Papua, Indonesia [31], and further emphasizes the need to test neonates born of positive mothers.

## Limitations

This was an analytical cross-sectional study. Although efforts were made to control for confounders at the design and analysis stages, the lack of data on the identified confounders such as nutritional status, other comorbidities, and other unidentified confounders may have affected the observations.

Reliance on patients' recall with regards to the administration of IPT through prescription (not DOT) and use of ITNs was another limitation to this study. To some extent evidence of prescription from the patients' records confirmed IPT utilization, unlike those considered as users based on the report of having slept under an ITN the night before the study. There was also quite a small number of mothers who reported receiving IPTp by DOT hence, adequate comparisons of the method of administration and the delivery outcome were not possible.

A significant number of women who deliver outside health facilities in this region may have had adverse birth outcomes. However, this study had the limitation of recruiting participants at a delivery unit.

## Conclusion

The effective use of malaria preventive strategies (IPT-SP and ITN) is generally low. Most of the women took less than three doses and there was no strict adherence to the recommended DOT. The prevalence of Plasmodium infection during pregnancy in our study was low though maternal anaemia and LBW were higher than other studies have documented in Eastern Uganda. Also, mRDT may be more reliable for the detection of malaria in pregnancy than slide microscopy.

## Recommendations

1. Efforts are needed to sensitize mothers to attend ANC as per the current WHO recommendation and scale up the uptake of optimal doses of IPT-SP and ensuring adherence to DOT- based administration of IPT by the health care workers.

2. A longitudinal study may be required to evaluate the relationship between the uptake of re commended doses of IPT-SP and delivery outcomes.

## Supporting information

**S1 Data.**
(DTA)

## Acknowledgments

With great pleasure, we acknowledge all our study participants who gave very useful information and provided samples for this study. Thanks to the staff of the Department of Obstetrics and Gynaecology-Mbale Regional Referral Hospital, who created a conducive environment to carry out this study. Our gratitude goes to the staff of the Department of Microbiology and immunology, faculty of Health Sciences, Busitema University, for the support of this study.

## Author Contributions

**Conceptualization:** Rebecca Nekaka, Jacob Stanley Iramiot.

**Data curation:** Rebecca Nekaka, Julius Nteziyaremye, Paul Oboth, Jacob Stanley Iramiot, Julius Wandabwa.

**Formal analysis:** Rebecca Nekaka.

**Funding acquisition:** Rebecca Nekaka.

**Investigation:** Rebecca Nekaka.

**Methodology:** Rebecca Nekaka, Jacob Stanley Iramiot.

**Supervision:** Jacob Stanley Iramiot, Julius Wandabwa.

**Writing – original draft:** Rebecca Nekaka.

**Writing – review & editing:** Julius Nteziyaremye, Paul Oboth, Jacob Stanley Iramiot, Julius Wandabwa.

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
