## [Decision Letter · Decision Letter 0]

18 Jun 2020

PONE-D-20-13366

Malaria Preventive Practices and Delivery Outcomes: A Cross-sectional Study of Parturient Women in a Tertiary Hospital in Eastern Uganda.

PLOS ONE

Dear Mr Iramiot

Thank you for submitting your manuscript to PLOS ONE. After careful consideration, we feel that it has merit but does not fully meet PLOS ONE’s publication criteria as it currently stands. Therefore, we invite you to submit a revised version of the manuscript that addresses the points raised during the review process.

Please submit your revised manuscript by  15th August 2020. If you will need more time than this to complete your revisions, please reply to this message or contact the journal office at plosone@plos.org. Please include the following items when submitting your revised manuscript:

We look forward to receiving your revised manuscript.

Kind regards,

Professor Kwasi Torpey, MD PhD MPH

Academic Editor

PLOS ONE

Journal Requirements:

2. Please amend your list of authors on the manuscript to ensure that each author is linked to an affiliation. Authors’ affiliations should reflect the institution where the work was done (if authors moved subsequently, you can also list the new affiliation stating “current affiliation:….” as necessary).

3. Your ethics statement must appear in the Methods section of your manuscript. If your ethics statement is written in any section besides the Methods, please move it to the Methods section and delete it from any other section. Please also ensure that your ethics statement is included in your manuscript, as the ethics section of your online submission will not be published alongside your manuscript.

Reviewers' comments:

Reviewer's Responses to Questions

**Comments to the Author**

1. Is the manuscript technically sound, and do the data support the conclusions?

Reviewer #1: Yes

Reviewer #2: Yes

2. Has the statistical analysis been performed appropriately and rigorously? 

Reviewer #1: Yes

Reviewer #2: I Don't Know

3. Have the authors made all data underlying the findings in their manuscript fully available?

Reviewer #1: Yes

Reviewer #2: Yes

4. Is the manuscript presented in an intelligible fashion and written in standard English?

Reviewer #1: No

Reviewer #2: Yes

5. Review Comments to the Author

Reviewer #1: Major Comment

1. Almost all references in background part are wrong. The references mentioned are not providing the stated concept. Some are directly copied from the background of other studies, with no change in reference.

Dear authors, please make sure that you are using a reference because of the finding/conclusion of that work, not because of the background they used.

Minor comments

1. The background in abstract is not sufficient, it says nothing about the case in pregnant women, except what is mentioned as objective

2. If possible, avoid abbreviations in abstract or use in the expanded form in their first appearances eg. ANC

3. It is better to use country or city as keyword for easy searchability of the work. Eg. Uganda

4. Exclusion: it doesn’t give sense to exclude deliveries on weekends and public holidays in a research(specially funded one). Rationalize as this exclusion may rise the idea of bias. What was your reason?

5. Grammar and spelling errors: it needs language editing.

Examples: His could be….. instead of This could be….. 5th line of discussion

Higher that others…… instead of Higher than others….last line of conclusion

6. Conclusion: in the statement saying the effective use of malaria preventive strategies (IPT-SP and ITN) is generally still low, please do not use generally and still together. Use either of them. I recommend is generally low.

Reviewer #2: The study is interesting and noteworthy, and contributes to the needed body of evidence about access to and impact of malaria in pregnancy interventions in sub-Saharan Africa. That said, the rigor of the IPTp coverage data is ambiguous. They refer to women who did vs did not receive X doses of IPTp with SP, while simultaneously noting that over 85% received SP by prescription, not by DOT. The researchers do not meaningfully address the rigor of self-reported IPTp administration. The data is analyzed comparing women receiving vs not receiving IPTp, however there should be some acknowledgement that there may be differences between women who received IPTp by DOT, and that there may be differences in outcome between women who received IPT via DOT vs not (noting that if IPTp is not administered via DOT, it cannot be confirmed as taken). I recognize that it would be difficult to disaggregate the IPTp data for analysis by DOT vs non-DOT, as the DOT group is likely too small to produce significant results, however the data as presented may mislead readers about the relationship between IPTp and anemia, parasitemia and birth outcome.

There are several other minor revisions that should be made, such as updating the WHO recommended number of ANC visits to be in line with current guidelines.

I propose that the recommendation “Sensitizing women about IPT during health talks will encourage women to demand for DOT mode of IPT.” should be revised to focus on changing provider behavior. Providers are responsible for the care they offer and it is not reasonable to assume that pregnant women are empowered enough and have adequate agency to demand this service. Additionally, the differences in detection rates of parasitemia with smears vs RDT should be considered for a potentially recommendation.

The paragraph beginning “Of the 19 mothers with positive malaria Rapid Diagnostic Test (MRDT) at delivery” could be revised for clarity between the groups.

Several other places could benefit from revisions for clarity including “The women who took 2

or more doses had a higher likelihood of being compared to those who had less than two doses” and “Studies done by The difference in the prevalence between microscopy and RDT…”

6. PLOS authors have the option to publish the peer review history of their article (what does this mean?). If published, this will include your full peer review and any attached files.

Reviewer #1: No

Reviewer #2: No

---

## [Author Response · Author response to Decision Letter 0]

15 Jul 2020

Response to reviewers 

SN Comment Response 

Reviewer 1

Major comments

1 Almost all references in background part are wrong. The references mentioned are not providing the stated concept. Some are directly copied from the background of other studies, with no change in reference. 

Dear authors, please make sure that you are using a reference because of the finding/conclusion of that work, not because of the background they used All references have been reviewed and well aligned as recommended by the reviewer. Thank you

Minor comments

1 The background in abstract is not sufficient, it says nothing about the case in pregnant women, except what is mentioned as objective Corrections have been made as recommended by the reviewer.

2 If possible, avoid abbreviations in abstract or use in the expanded form in their first appearances eg. ANC Amended as recommended by the reviewer

3 It is better to use country or city as keyword for easy searchability of the work. Eg. Uganda Thank you for the guidance. This has been amended. 

4 Exclusion: it doesn’t give sense to exclude deliveries on weekends and public holidays in a research (specially funded one). Rationalize as this exclusion may rise the idea of bias. What was your reason? This has been corrected. The recruitment was actually done throughout the week.

5 Grammar and spelling errors: it needs language editing. 

Examples: His could be….. instead of This could be….. 5th line of discussion

 Higher that others…… instead of Higher than others….last line of conclusion Critical language editing has been done throughout the manuscript. Thank you.

6 Conclusion: in the statement saying the effective use of malaria preventive strategies (IPT-SP and ITN) is generally still low, please do not use generally and still together. Use either of them. I recommend is generally low.

 This has been corrected as recommended by the reviewer.

Reviewer 2

1 The study is interesting and noteworthy, and contributes to the needed body of evidence about access to and impact of malaria in pregnancy interventions in sub-Saharan Africa. That said, the rigor of the IPTp coverage data is ambiguous. They refer to women who did vs did not receive X doses of IPTp with SP, while simultaneously noting that over 85% received SP by prescription, not by DOT. The researchers do not meaningfully address the rigor of self-reported IPTp administration. Thank you. Amendments have been made for clarity. Self-reported IPTp administration has been acknowledged as a limitation to the study. 

2 The data is analyzed comparing women receiving vs not receiving IPTp, however there should be some acknowledgement that there may be differences between women who received IPTp by DOT, and that there may be differences in outcome between women who received IPT via DOT vs not (noting that if IPTp is not administered via DOT, it cannot be confirmed as taken). I recognize that it would be difficult to disaggregate the IPTp data for analysis by DOT vs non-DOT, as the DOT group is likely too small to produce significant results, however the data as presented may mislead readers about the relationship between IPTp and anemia, parasitemia and birth outcome. The authors have acknowledged this as a limitation to the study

3 There are several other minor revisions that should be made, such as updating the WHO recommended number of ANC visits to be in line with current guidelines.

I propose that the recommendation “Sensitizing women about IPT during health talks will encourage women to demand for DOT mode of IPT.” should be revised to focus on changing provider behavior. Providers are responsible for the care they offer and it is not reasonable to assume that pregnant women are empowered enough and have adequate agency to demand this service. Amendments have been made as recommended by the reviewer.

4 Additionally, the differences in detection rates of parasitemia with smears vs RDT should be considered for a potentially recommendation. This has been improved in the discussion section and also a has been drawn from our findings and from previous studies. 

5 The paragraph beginning “Of the 19 mothers with positive malaria Rapid Diagnostic Test (MRDT) at delivery” could be revised for clarity between the groups. This has been revised as advised by the reviewer.

6 Several other places could benefit from revisions for clarity including “The women who took 2 or more doses had a higher likelihood of being compared to those who had less than two doses” and “Studies done by The difference in the prevalence between microscopy and RDT…” The authors have recommended a longitudinal study to related IPTp uptake to the delivery outcome

---

## [Editor Report · Decision Letter 1]

20 Jul 2020

PONE-D-20-13366R1

Malaria Preventive Practices and Delivery Outcomes: A Cross-sectional Study of Parturient Women in a Tertiary Hospital in Eastern Uganda.

PLOS ONE

Dear Mr Iramiot,

Thank you for submitting your manuscript to PLOS ONE. After careful consideration, we feel that it has merit but does not fully meet PLOS ONE’s publication criteria as it currently stands. Therefore, we invite you to submit a revised version of the manuscript that addresses the points raised during the review process.

The manuscript needs significant copyediting and language corrections before it can be published. There are several examples of grammatical errors and inappropriate capitalization in the manuscript. Using the abstract as an example the following errors are noted

1. long term effects consequences

2. Inappropriate capitalization of antenatal in the middle of a sentence

3. We aimed to determine the use...…..    This is could be simply We determined

4. Typo Isectides

There are several more errors in the manuscript and I strongly recommend a native speaker copyedits and c

We look forward to receiving your revised manuscript.

Kind regards,

Professor Kwasi Torpey, MD PhD MPH

Academic Editor

PLOS ONE

---

## [Author Response · Author response to Decision Letter 1]

23 Jul 2020

Response to Reviewers 

SN Comment Response

1 Please remove any funding-related text from the manuscript and let us know how you would like to update your Funding Statement. Amendments have been made as advised by the reviewer

---

## [Editor Report · Decision Letter 2]

27 Jul 2020

Malaria Preventive Practices and Delivery Outcomes: A Cross-sectional Study of Parturient Women in a Tertiary Hospital in Eastern Uganda.

PONE-D-20-13366R2

Dear Mr Iramiot,

We’re pleased to inform you that your manuscript has been judged scientifically suitable for publication and will be formally accepted for publication once it meets all outstanding technical requirements.

Kind regards,

Professor Kwasi Torpey, MD PhD MPH

Academic Editor

PLOS ONE
---

## [Editor Report · Acceptance letter]

29 Jul 2020

PONE-D-20-13366R2 

Malaria Preventive Practices and Delivery Outcomes: A Cross-sectional Study of Parturient Women in a Tertiary Hospital in Eastern Uganda. 

Dear Dr. Iramiot:

I'm pleased to inform you that your manuscript has been deemed suitable for publication in PLOS ONE. Congratulations! Your manuscript is now with our production department. 

Kind regards, 

on behalf of

Professor Kwasi Torpey 

Academic Editor

PLOS ONE